# Computer Vision-Based Classification of Flow Regime and Vapor Quality in Vertical Two-Phase Flow

**DOI:** 10.3390/s22030996

**Published:** 2022-01-27

**Authors:** Shai Kadish, David Schmid, Jarryd Son, Edward Boje

**Affiliations:** 1Department of Electrical Engineering, University of Cape Town, Cape Town 8001, South Africa; jarryd.son@uct.ac.za (J.S.); edward.boje@uct.ac.za (E.B.); 2The European Organization for Nuclear Research (CERN), 1211 Meyrin, Switzerland; david.schmid@cern.ch

**Keywords:** flow regime, vapor quality, computer vision, machine learning

## Abstract

This paper presents a method to classify flow regime and vapor quality in vertical two-phase (vapor-liquid) flow, using a video of the flow as the input; this represents the first high-performing and entirely camera image-based method for the classification of a vertical flow regime (which is effective across a wide range of regimes) and the first image-based tool for estimating vapor quality. The approach makes use of computer vision techniques and deep learning to train a convolutional neural network (CNN), which is used for individual frame classification and image feature extraction, and a deep long short-term memory (LSTM) network, used to capture temporal information present in a sequence of image feature sets and to make a final vapor quality or flow regime classification. This novel architecture for two-phase flow studies achieves accurate flow regime and vapor quality classifications in a practical application to two-phase CO_2_ flow in vertical tubes, based on offline data and an online prototype implementation, developed as a proof of concept for the use of these models within a feedback control loop. The use of automatically selected image features, produced by a CNN architecture in three distinct tasks comprising flow-image classification, flow-regime classification, and vapor quality prediction, confirms that these features are robust and useful, and offer a viable alternative to manually extracting image features for image-based flow studies. The successful application of the LSTM network reveals the significance of temporal information for image-based studies of two-phase flow.

## 1. Introduction

A flow regime describes the spatial distribution between the vapor and liquid phases in a two-phase flow, with the different regimes being identified by the gas bubble characteristics inherent to each regime [1]. The classification of flow regimes for multiphase flow is essential in many industrial sectors, such as the energy, metallurgical, and processing industries. The study of flow regime is important because it reveals essential information about flow behavior, as well as the physical flow parameters of the two-phase flow under investigation [2]. Different flow regimes can be observed across different flow channel shapes, orientations, and operating conditions, with different flow regimes also arising because of properties of the flow itself, including phase velocity and vapor quality [2]. Much of the literature aims to relate flow regime classes to the measured physical characteristics of the flow and to define the regimes and their transitions in terms of these physical characteristics, for a given set of experimental conditions [2,3,4,5].

The physical parameters or features used in the classification of flow regimes are classically extracted from the flow by direct measurement, using a variety of instruments [6,7,8,9,10,11,12,13,14,15]. Some of these instruments require direct contact with the flow [6,7,8,9,10], while other methods read flow data in a non-intrusive manner [11,12,13,14,15]. Image-processing techniques represent a non-intrusive method by which to extract flow features for the purpose of flow regime study. In most incarnations of these methodologies, the useful image features are selected and extracted manually [16,17,18,19,20,21,22,23,24], with any flow classification being made in terms of groupings of these features.

Modern computer-vision architectures allow for the automatic extraction and utilization of image features to solve classification problems. These image-based classifiers are implemented most using convolutional neural networks (CNNs) within their architectures [25,26,27]. CNNs have been implemented to analyze two-phase flow in some limited cases; they have been used to study bubbly flow [28,29] and as a tool for the analysis of complex flow characteristics [30], but the use of these networks as image-based classifiers of flow regime has been limited. Branston et al. [31] achieved good results in image-based vertical flow regime classification by using a CNN to classify cross-sectional images of the flow channel, produced by an advanced wire-mesh sensor. A study by Du et al. [32] compared three CNN-based architectures for their ability to classify the vertical flow regime of a given input image, captured from the observation section of a flow channel using a camera. Seal et al. [33] also achieved good results in the image-based classification of vertical flow across four vertical regimes using a CNN; however, their study made use of a small dataset of under 4000 images, which most likely led to biases in the data. Although both Du et al. [32] and Seal et al. [33] trained classifiers using a small number of visually distinct flow regime classes, the successes in their results point to the possibility of more complex flow regime classification using CNNs.

The novel contributions found in this work, and their corresponding benefits, include:The development of a high-performing CNN-based flow regime classifier for vertical flow, which is applicable to a wide range of flow regimes (with some being visually similar); the classifier is trained using a large dataset, one where the only inputs are images captured by a camera.The detailing of the first published deep learning and image-based method (with mass flow rate and pressure also included as inputs) for vapor quality estimation.The fact that these methods make use of only camera images (and static flow parameters for vapor quality estimation) leads to them being more accessible, as they require less technical or domain-specific knowledge for deployment.The use of only camera images also allows for the real-time deployment of these classifiers. The real-time deployment (at a prototype level) of a flow regime and vapor quality classifier is a novel achievement that is presented in this paper. These real-time implementations are advantageous in that they allow the above models to be utilized within a control feedback loop.Both these methods have CNN + LSTM [34,35,36,37] architectures, allowing them to make use of a sequence of images, rather than a single one, when making a classification. This design has not been applied to the study of multi-phase flow before.The LSTM’s use of image sequences to account for temporal flow characteristics will be shown to be useful in image-based two-phase flow studies.By utilizing image features extracted by a CNN network for these distinct tasks, this method is shown to be a viable alternative to manual image feature extraction in analyzing two-phase flow.

In Section 2.1, the experimental environment used to generate two-phase flow video recordings and real-time data is described. Section 2.2 provides a detailed description of the model architectures used. In Section 2.3, the data and classes used for the training of the models are defined, as well as the training process itself. The results are presented for offline and real-time testing in Section 3, followed by a discussion in Section 4 and a conclusion in Section 5.

## 2. Materials and Methods

### 2.1. Experimental Setup

#### 2.1.1. Two-Phase Flow Data Generation

The details of the experimental rig used to generate the two-phase flow can be found in Schmid et al. [3]. For the results presented here, two-phase fluid is recorded, flowing in an upward direction through an 8 mm-diameter circular channel to generate the video data. For each set of video data captured, constant mass flow rates and fluid-specific enthalpies, ranging from 0.005 to 0.023 kg/s—resulting in mass velocities in the order of 100 to 450 kg m^−2^ s^−1^ within the test rig used—and 115 to 445 kJ/kg, respectively, were used. The pressure of the channel ranged from 1683 to 3970 kPa. The combination of mass flow rates and specific enthalpies at the inlet of the test section produced video data representing the full range of possible vapor qualities (from 0 to 1) and flow regimes in the test section.

#### 2.1.2. Image Acquisition for Offline Testing and Model Training

To capture flow footage for initial off-line evaluation and for use as model training data, a Photron FastCam Mini AX100 camera was used. Images were captured at a resolution of 512 × 512 pixels.

For the flow-regime classifier and vapor quality estimator, videos were recorded at frame rates of 30 FPS and 10 FPS respectively, with these values arising as the optimal frame rates for their specific tasks from a hyperparameter search. The results of this search for the flow-regime classifier are depicted in Figure 1. Lower frame rates allow a longer video clip in the temporal dimension to be processed over a smaller number of frames, but if these rates are too low (as seen for the flow regime classifier, at rates below 20 FPS in Figure 1), temporal aliasing might make it difficult for the LSTM network to extract useful temporal features between frames. If the frame rate is too high (as is the case for the regime classifier for rates above 40 FPS), the time between frames is too short for useful temporal features to be learned by the network.

Shutter speed for this camera presented no constraint in terms of minimizing motion artifacts within the videos. Consistent backlighting was applied to the test section to retain consistency in the image intensity, seen across the video data captured.

#### 2.1.3. Laboratory Prototype and Real-Time Image Acquisition

A laboratory prototype for real-time data acquisition and processing was developed using a Jetson Nano micro-computer with an ArduCam OV9281 camera, as shown in Figure 2c. The Jetson Nano has a 128-core GPU, allowing it to process images (using the architectures described in Section 2.2) at frame rates that are acceptable to demonstrate the principle at low cost. The data captured during real-time testing was generated in the same experimental environment as the offline data.

### 2.2. Model Architecture

A key aspect of the architecture is its use of separate classes for image frames and frame sequences. It was observed that images with highly similar features occur across multiple flow regimes. When training a network that classifies frames based upon flow regime classes, the inter-class confusion is high because of these similarities. It is also observed that an individual’s perception and definition of flow regime is not only characterized by the spatial distribution of the gas and liquid within the two-phase flow but also by the way in which it changes over time. For instance, a transitional flow regime might appear distinctly as one flow regime in a single frame, but it is defined as transitional on the basis of how the flow behaves across multiple frames. This means that temporal information must be key for image-based flow regime classification. Combining these two observations, a novel method to achieve flow regime classification is developed, where a sequence of frame features is used to predict the flow regime. Similar network architectures have been implemented in other tasks related to image sequence analysis [35,36,37].

The network architecture of the flow regime classifier (FlowNet), as described in Figure 3, is made up of three components, with the data path for the network flowing downwards. The input to the system is an image from a sequence of frames, and the output is a flow regime class prediction. A vapor quality regression network (VaporNet) is implemented by adjusting the classification layers of FlowNet.

The first component of the system is the image feature-extraction component. This operation is performed by the frame classifier (FrameNet). Each image in the sequence is passed through FrameNet to extract the image features for further processing. This component of the network transforms individual image frames into a set of image features, as seen in Figure 4. A sequence of image feature sets is used as the input to the recurrent layer, which is made up of LSTM units. These LSTM units are utilized to capture temporal information across the feature sequence. The data at the output of the recurrent layer is then passed into a set of dense layers making up the classification layer, which, in turn, outputs a final classification (for FlowNet) or regression value (for VaporNet). These three components are further detailed below.

#### 2.2.1. Image Feature Extraction

The architecture selected for the frame classifier was that of ResNet101 [25]. This architecture was chosen for its state-of-the-art classification accuracy. ResNet101 was modified to have an output layer of seven nodes, so as to correspond to the frame classes for which it is trained. The features extracted from FrameNet for the other two models are obtained from the 2048 inputs to the dense output layer of FrameNet. These features represent a reduction in the image data to a set of sparse features that are useful for flow-image classification. These feature sets are fed directly into a dense layer for classification within their FrameNet classes or, in the case of VaporNet and FlowNet, passed into the recurrent layer to be analyzed within a sequence of feature sets.

#### 2.2.2. Recurrent Layer

The analysis of the sequences of feature sets, with each element of a sequence being made up of 2048 features corresponding to an input frame, is performed by LSTM units [34]. LSTMs are a form of Recurrent Neural Network (RNN) [38]. RNNs can make predictions based upon past contextual information, allowing them to interpret temporal information that is present in a sequence. This layer makes use of two stacked LSTM networks to form a deep LSTM; this structure allows for greater levels of abstraction in terms of the temporal input information to the LSTM [39]. The chosen deep LSTM depth of two was determined through optimization by hyperparameter search. For FlowNet, the sequence length is 20 frames, while for VaporNet, this sequence is 50 frames long. The LSTM units have a hidden state (a vector updated at each timestep, which encodes past contextual information) of 256 nodes in length, a value which, along with the sequence lengths, was determined through a hyperparameter search. The hidden state of the final LSTM unit in the second layer of the deep LSTM network is then passed into the classification layer. This state is informed by past information, derived from the outputs of the previous LSTM units, which are used to analyze earlier frames. Although bidirectional LSTMs have been shown to produce better results than unidirectional LSTMs for similar problems [35,40], a unidirectional design allows for the possibility of real-time sequence analysis.

#### 2.2.3. Classification Layer

The features extracted in the previous layers are processed by a set of dense layers to make a final prediction. FlowNet uses one dense layer made up of 256 nodes, which takes the output from the LSTM network at its input and outputs to eight nodes with *soft-max* activation functions (one node for each class of flow regime). The *soft-max* function was selected because it is an appropriate output function for use with the *cross-entropy* loss function employed during classification training. The first of VaporNet’s two dense layers contains one node that takes in the hidden state from the recurrent layer. The mass flow rate and saturation temperature (which relate to the pressure of the system), used in generating the sequence of frames, are passed into the second dense layer, containing 256 units, along with the output from the single node. The second dense layer connects to a single node, with a linear activation function to output the final vapor quality prediction. The number of units and depth of the classification layer were determined through a hyperparameter search.

### 2.3. Class Definitions and Data Preparation

The flow regime classes used to classify a sequence of frames were those found in the literature for vertical up-flow [41]. In addition to these classic regimes, classes that represent transitional zones between these regimes are included to achieve a higher resolution in the regime classification. The flow regime classes used to classify sequences of frames in order of ascending vapor quality for fixed flow conditions, with all transitional classes being hyphenated, are as follows: liquid-bubbly, bubbly, bubbly-slug, slug, slug-churn, churn, annular and mist flow.

FrameNet is trained to classify two-phase flow images within image classes that exist across multiple flow regimes (for instance, the first four classes listed below would be common in bubbly and slug regimes but might be observed transiently in other regimes). By training the network for this task, a robust and well-generalized set of image features (for two-phase flow images) is encoded within the convolutional layers of FrameNet, with these features also being useful for FlowNet and VaporNet. The flow image classes, used to define individual frames of the flow, can be determined through the qualitative analysis of an image set that includes frames from video clips of each flow regime. From this analysis, frame classes are shown in Figure 5, and can be described as follows:Liquid/tiny bubbles: A small number of tiny, discrete gas bubbles flow in a continuous liquid phase.Small bubbles: A few small, discrete gas bubbles flow in a continuous liquid phase.Big bubbles: A few small, discrete gas bubbles, with some large spherical bubbles and slug-like bubbles within the fluid, flow in a continuous liquid phase.Dense bubbles/Taylor bubbles: Many small- to medium-sized discrete bubbles flow in a continuous liquid phase. The bubbles are distributed more consistently and densely across the image, with more than half the viewing section of the tube being taken up by bubbles. Taylor bubbles are also found in this frame class.Churn: A mix of gas and liquid that flows chaotically, with no visible bubbles.Annular: A gas core forms from the center of the pipe. A wavy liquid film flows along the walls of the pipe, and liquid droplets are dispersed within the gas core.Mist/vapor: No liquid is visible, as a continuous gas phase flows through the channel.

A balanced dataset made up of 39,261 frames, labeled with the above frame classes, was generated by extracting and manually classifying a set of image frames from the videos of two-phase flow, generated by the test rig and used for training FrameNet. These images were standardized and cropped in such a way that the edges of the flow channel are on the borders of the image. The images are then resized to 256 × 256 pixels, as is required for inputs to FrameNet [25].

A second dataset used for the training, validating, and offline testing of FlowNet was generated by using FrameNet to extract features for each frame in a 20-frame sequence. These 20-frame clips are extracted from flow videos generated by the test rig, captured at 30 FPS. Each data point for FlowNet is therefore a sequence of 20 feature sets, corresponding to the 20-frame clip that is being classified in terms of its flow regime. The label of each clip is assigned based on the flow regime label of the video from which it is partitioned.

A third dataset is required for VaporNet, also made up of image features extracted by FrameNet. Features from 50-frame clips were used (from video captured at 10 FPS), with each clip being labeled for training with an experimentally derived vapor-quality value, describing the video from which the clip was taken. Additionally, VaporNet requires information on the saturation temperature and mass flow rate of the flow at the time that the clips were taken as two additional input channels. Both this dataset and FlowNet’s dataset are balanced and are made up of 35,232 data points.

The use of balanced datasets for all models ensures that the networks learn data characteristics that are well-generalized across the training data. Additionally, by using balanced datasets, poorly classified classes are highlighted as having difficult features or characteristics for the model to learn, rather than as having an insufficient training subset. This response is true because if the subset size is sufficient for a model to accurately learn the features of another class, poor performance is not a result of the sample size but rather a result of the specific features of the poorly classified class.

### 2.4. Model Training

Each of the three neural networks was trained using the methods of gradient descent and backpropagation. A *cross-entropy* loss function was used for training, with this loss function selected because of its excellent performance in classification tasks [42], and a stochastic gradient descent (SGD) optimizer was employed because of its good generalization [43]. A *cross-entropy* loss function was used for FlowNet, but an *Adam* [44] optimization function was deployed for faster convergence. For VaporNet, a Mean Square Error (MSE) loss function was implemented and optimized using *Adam*. The detailed hyperparameters used for training can be seen in Table 1.

FrameNet was trained first, separately from the other two networks, as it is used to generate the training and offline testing data for the other two networks. The recurrent and classification layers of FlowNet and VaporNet were then trained using sets of this feature data generated by FrameNet.

Training of the models was carried out using a workstation with a 2.3 GHz Intel^®^ Xeon^®^ E5-2630 CPU, 128 GB RAM, and an NVIDIA^®^ Kepler™ K40 M GPU with 12 GB of GPU accelerator memory. All networks were implemented using the *PyTorch* framework [45]. A FrameNet model takes approximately 4 h to train, with VaporNet and FlowNet models being trained in under one hour.

## 3. Results

### 3.1. Model Performance

To evaluate the offline performance of the three network types on their respective datasets, k-fold cross-validation [46,47] was implemented for each network. This method was implemented in lieu of the option to test the networks on multiple datasets. This method provides an estimate of a model’s prediction performance and sensitivity to variations in training data. *k* was chosen to be 5, as 5-fold cross-validation has been shown to offer an acceptable balance between bias and variance in its results [47]. By performing 5-fold cross-validation, five unique sets of model parameters are produced for each of the three network types, with the performance of the networks being evaluated based on the results across their models. Network-specific test sets (generated using the FastCam) that were not used for the training of the models were used to evaluate the performance of each model. Testing the models on unseen data ensures that the results reflect how the model might perform in practical applications; by using the same test set across the five models of the same network type, an objective comparison can be made. Each of the test sets for the three respective models was made up of 26,200 data points.

The results from the cross-validation of each network can be seen in Table 2. While the columns describing FlowNet and FrameNet show the accuracy of the classifiers (the percentage of classifications that were correct out of the test set), the column describing VaporNet makes use of the root mean square error (RMSE). This value is a measure of the average error made by the model in terms of vapor quality prediction. Figure 6, Figure 7a and Figure 8a were generated using the highest-performing model for each respective network.

Figure 6 and Figure 7a show the normalized confusion matrices for FrameNet and FlowNet, respectively. Confusion matrices reveal how a classification model performs across its classes, revealing any prediction biases that a model might have.

Figure 8a shows a boxplot for the set of predictions made for each vapor quality value being tested. These data points were generated across a range of mass flow rates and pressures. Because each vapor quality being tested is represented by a video with a respective average vapor quality, there are multiple vapor quality predictions made for each video, achieved by using clips from the larger video for predictions. Each set of predictions is represented by a boxplot in Figure 8a, with a tighter distribution of predictions representing more confident predictions for that vapor quality.

### 3.2. Real-Time Performance

Performance testing revealed that the constraint on interpolation speeds is from the *CNN* frame classifier, with the *LSTM* network generating outputs orders of magnitude faster. This testing also revealed that the Jetson Nano could process individual frames using FrameNet (which has ResNet101 architecture) at a stable frame rate of 10 FPS. While this interpolation speed is fast enough for VaporNet, FlowNet was trained using data captured at 30 FPS. It was found that the Jetson Nano can process frames using ResNet18 [25] (a version of ResNet with fewer parameters and, therefore, a lower memory requirement) at 20 FPS, and FlowNet and FrameNet were retrained using this new architecture, with results as shown in Figure 1.

This figure shows that FlowNet’s performance when using ResNet18 at 20 FPS is comparable to its performance when using ResNet101 at 30 FPS (with mean model accuracies of 89.4 and 91.7%, respectively), and, as such, the real-time deployment of FlowNet uses ResNet18 architecture at 20 FPS. The confusion matrix produced by this model, while processing real-time data, can be seen in Figure 7b.

Because the Jetson Nano can run the optimal version of VaporNet, no changes are made to this model for real-time implementation. The results produced by the real-time implementation of VaporNet on the laboratory prototype can be seen in Figure 8b.

## 4. Discussion

### 4.1. Offline Results

Table 2 shows that the accuracy of FrameNet is high across its models. Although the prediction accuracies of FrameNet are mostly consistent across classes, there are certain frame classes that produce a disproportionate number of classification errors, with these misclassifications occurring mainly between neighboring classes, as seen in Figure 6. Most of these incorrect classifications occur between the Small Bubbles and Big Bubbles classes; this may be due to the similarity between the image features found in these classes, due to the similarity in the bubbles: both classes have nearly spherical bubbles with only minor deformations in their shapes. This issue of similar features being found in images existing in different classes might also extend to the classification of a sequence of images, potentially causing confusion between the churn and annular FlowNet classes. Another possible cause for the above confusion is the introduction of errors in the FrameNet data labels by the manual labeling of training and test data.

As seen in Table 2, both VaporNet and FlowNet have moderate standard deviation values when compared to FrameNet in their cross-validation results, which indicates that the performances of these networks are more sensitive to the data used for training and that the networks may not generalize as well to a real-world test. This could be from the increased model complexity introduced by the recurrent network layers. Another explanation is that the sequence data used to train these networks comprised labeled videos that were truncated and partitioned into sets of clips that share a label with their overall video. It was observed that varying flow regimes arose among clips from the same video, yet all clips shared the label of the overall video. This resulted in the incorrect labeling of some clips. This led to the network overfitting to incorrectly labeled clips, producing the noted variance in results. This also explains the incorrect FlowNet predictions between consecutive flow regimes, as seen in Figure 7a, because these instances would represent clips of the overall video where the flow regime transitions from the average regime of the video to a transitional regime in a class preceding or following the average regime class. The way to interpret the clip predictions would then be that, when accumulated, they express the flow regime makeup of the overall video. In all test cases, the average flow regime predicted across the set of clips in a given video was correct for that video.

Despite this, FlowNet achieved a mean accuracy of 92%, with almost all incorrect classifications falling into neighboring flow regimes. The results gave good confidence in predictions for the current application. The model architecture, therefore, has the capacity to be trained successfully for the current task, but there is a tendency to overfit the training data, producing a high variance in results across models, as discussed above.

Figure 8a shows a clear correlation between vapor quality predictions and true vapor quality labels, with generally accurate, high-confidence predictions being made, with the following exceptions: between the label ranges of 0.1 and 0.2, the network showed a wider range of predictions and, therefore, a lower confidence per label. This range of vapor qualities mostly produces slug flow, with videos of this flow regime transitioning between clips of liquid-bubbly, to bubbly-slug, to slug, to slug-churn. Because this vapor quality range produces videos that change significantly over time, it is not surprising that different average vapor qualities would be predicted across the set of clips for such a video. Nevertheless, the mean vapor quality predictions for these videos are still reasonably accurate. There is a distinct drop in accuracy in the range from 0.6 to 0.9. This region often represents annular and mist flows. The network’s tendency to under-predict vapor quality in this region could arise because the image features extracted by FrameNet for flow-image classification do not contain the information content required to accurately predict the vapor quality of high-vapor-quality flow. This explains the seemingly uniform classification of these videos, as seen in Figure 8a, as the image data required to calculate their vapor quality to a more accurate degree is lost by FrameNet, which does not make use of this data for flow image classification. This problem is an attractive subject for further research and could potentially be solved by retraining FrameNet (and VaporNet on these new FrameNet features) so that features specific to this vapor quality range are extracted, rather than the broader features used here for analyzing the entire range of vapor qualities.

Table 2 reveals that VaporNet can predict vapor quality with a mean RMSE of 5% of full scale, across the partitions of its training data. VaporNet can, therefore, be applied to applications where the accuracy required falls within this error range.

Figure 9 highlights the fact that classifying flow regime by sequence is an appropriate design decision: FrameNet identifies different image classes within the same flow regime, whereas FlowNet analyses the sequence of frames to make a correct classification on the entire set. This shows that temporal information is useful when classifying flow regimes from image data. Additionally, when observing this sequence of frames, the deficiencies of FrameNet as a flow regime classifier are highlighted; FrameNet classifies frames 1, 2, and 3 within different classes, yet they all belong to the same flow regime class. This shows that despite FrameNet achieving higher accuracy than FlowNet, it is fundamentally unsuitable for flow-regime classification because it does not account for temporal information.

### 4.2. Real-Time Results

The superiority of ResNet18 at 20 FPS over ResNet101 at 10 FPS (with these frame rates being the highest-performing for each model in terms of the Jetson Nano’s real-time capabilities) for FlowNet, as shown in Figure 1, confirms that frame rate is a crucial hyperparameter when designing a network for flow regime classification; this comparison reveals that the model with far fewer network parameters but a more optimal frame rate performs significantly better.

When comparing online and offline results for FlowNet (Figure 7a vs. Figure 7b) the greatest decrease in accuracy occurs in the case of the mist flow class. Because images of mist flow are very similar, it is likely that FlowNet makes use of image features that are highly specific to the lighting conditions found in the training data when classifying this regime. For the data collected using the real-time prototype, there are inherent differences in the input images when compared to the training data, due to the altered camera positioning and different camera parameters (such as aperture, white balance, shutter speed, resolution, etc.). This leads to a change in the features found within an image and, thus, affects the performance of the classifier for all classes, but it most negatively affects mist flow, which contains more consistent features across images. VaporNet displays a similar drop in performance at higher vapor levels, where mist flow is likely to occur, as seen in Figure 8b.

The decreased performances of both models during real-time testing, as seen when comparing Figure 7a,b and Figure 8a,b and from real-time performance metrics, reveal that FlowNet and VaporNet achieve 61% classification accuracy and an RMSE of 0.1, respectively (compared to values of 89.4% and 4.4 × 10^−2^, respectively, for their offline counterparts), are the results of the different sets of image features found between the real-time and training data, and not the results of the change to real-time data capturing and processing. The fact that the outputs from both models still follow a desirable trend, despite the differences among images between datasets, indicates that these models could be improved, through retraining with data more like those seen at the test-time, to achieve similar performances to those seen in their offline implementations.

## 5. Conclusions

FrameNet was successfully trained to classify individual images from a two-phase CO_2_ flow, but it was shown that individual frame classification can be misleading when considering flow with transient properties. VaporNet and FlowNet address this issue by incorporating temporal information through the addition of LSTM networks to their architectures, while making use of spatial information provided by FrameNet. The results produced by these networks, and the related qualitative arguments presented in this work, encourage the inclusion of temporal information in further image-based studies of two-phase flow. Additionally, the image feature detectors learned by the convolutional layers of FrameNet were shown to be useful for all three network types, confirming that these feature detectors are robust and produce a well-generalized reduction in the image information content, and that this approach is a viable alternative to manual feature extraction for the study of two-phase flow images. Encouraging results were also seen in the prototype real-time implementations of these models, with the possibility of improving these results through the retraining of the models using data more like that seen at test-time.

## Figures and Tables

**Figure 1 sensors-22-00996-f001:**
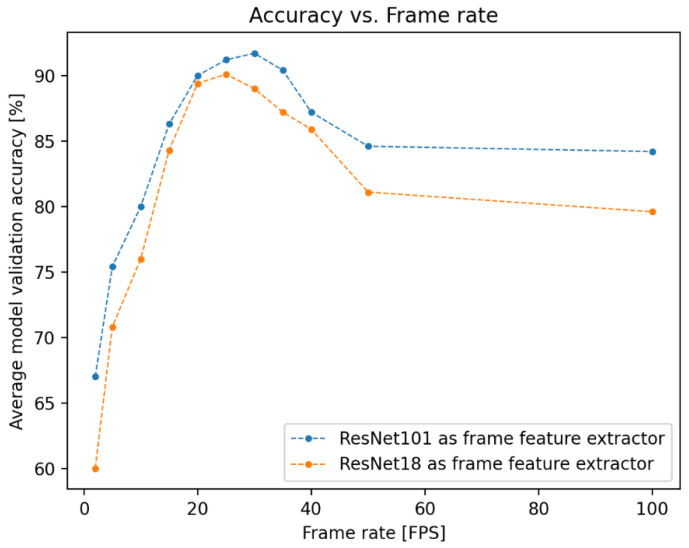
Accuracy vs. frame rate for the flow regime classifier (FlowNet). Depicted here are the results of the hyperparameter searches run during the training of the networks used for offline (using ResNet101) and online (using ResNet18) testing. These results are generated with offline data from the FastCam; as such, performance is expected to degrade in online testing, where data generated by the ArduCam is used.

**Figure 2 sensors-22-00996-f002:**
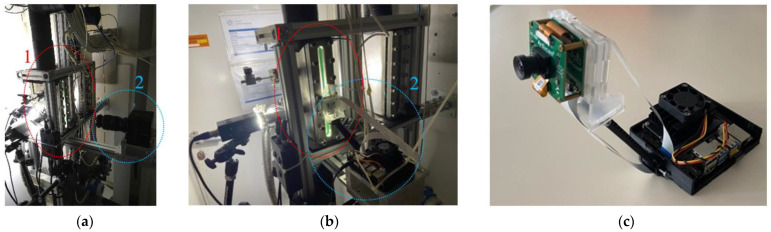
The experimental setup: (**a**) the FastCam (2) deployed to collect data from the observation section of the test rig (1) for training and offline testing; (**b**) The prototype (2) deployed to capture and process live video footage from the observation section of the test rig (1); (**c**) the prototype.

**Figure 3 sensors-22-00996-f003:**
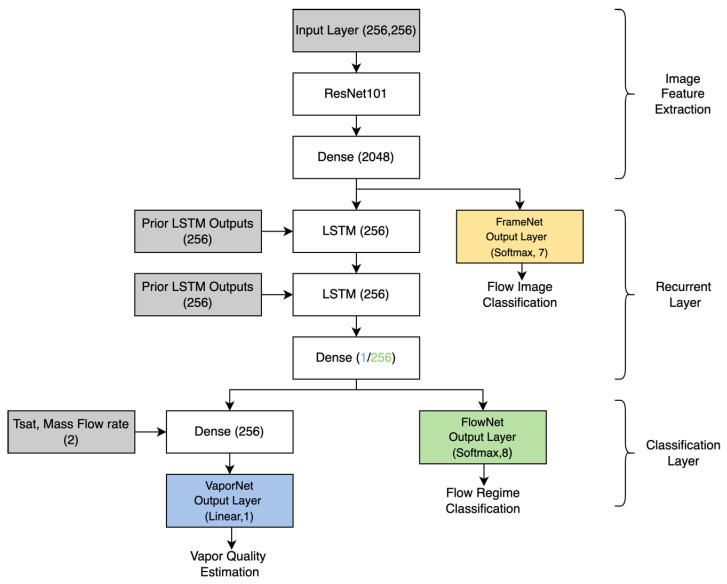
High-level network architecture, with paths for FrameNet (yellow), VaporNet (blue), and FlowNet (green). The number of neurons in each component is shown in brackets.

**Figure 4 sensors-22-00996-f004:**
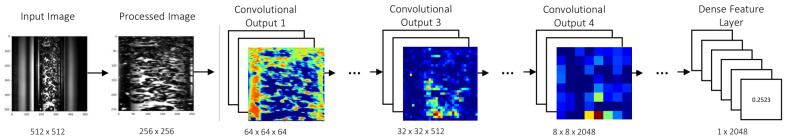
Image feature reduction: FrameNet passes an input image through a series of convolutional layers to identify the reduced image features in the form of a set of single values, useful for frame classification.

**Figure 5 sensors-22-00996-f005:**
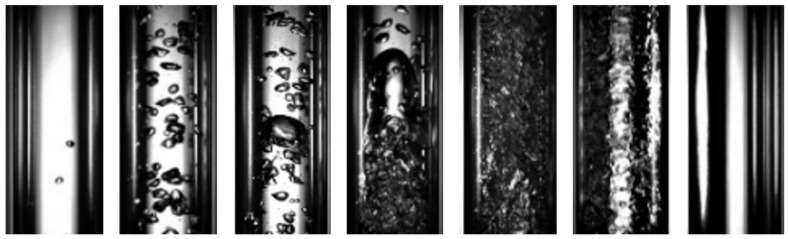
Frame classes (**left** to **right)**: liquid/tiny bubbles, small bubbles, big bubbles, dense bubbles/Taylor bubbles, churn, annular, and mist/vapor.

**Figure 6 sensors-22-00996-f006:**
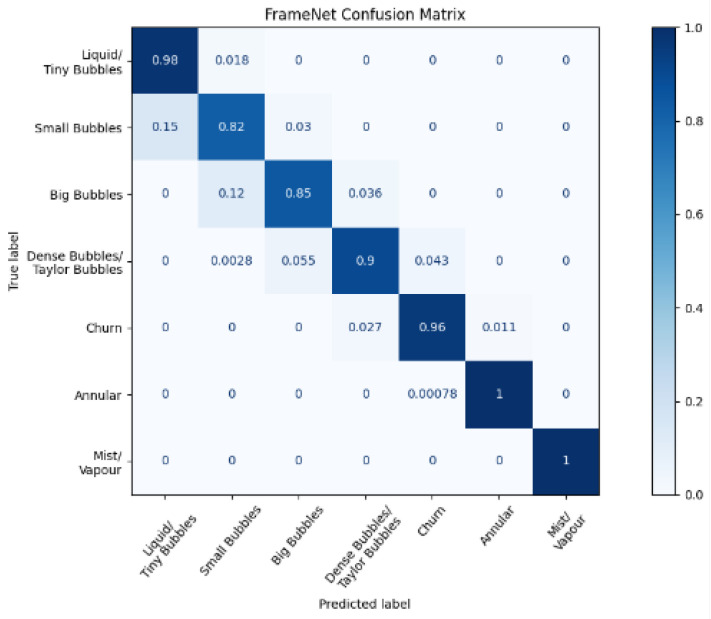
Normalized confusion matrix for FrameNet.

**Figure 7 sensors-22-00996-f007:**
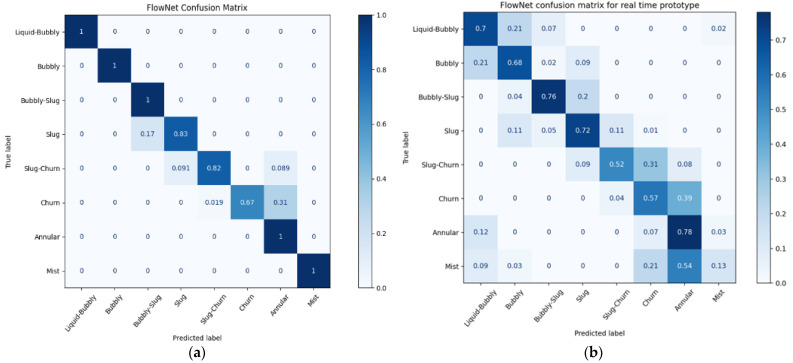
Normalized confusion matrices for FlowNet: (**a**) during offline testing; (**b**) during online testing using the experimental prototype.

**Figure 8 sensors-22-00996-f008:**
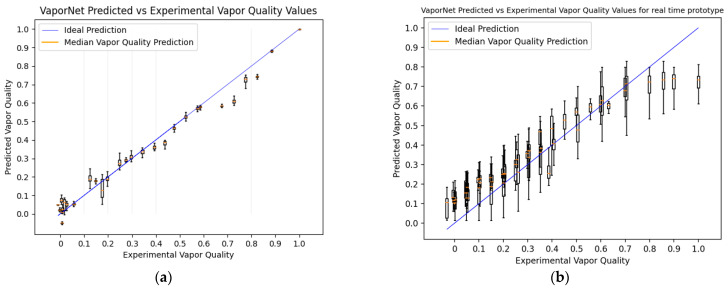
VaporNet predicted values vs. experimental values: (**a**) using offline data processing; (**b**) deployed on the laboratory prototype, collecting live data.

**Figure 9 sensors-22-00996-f009:**
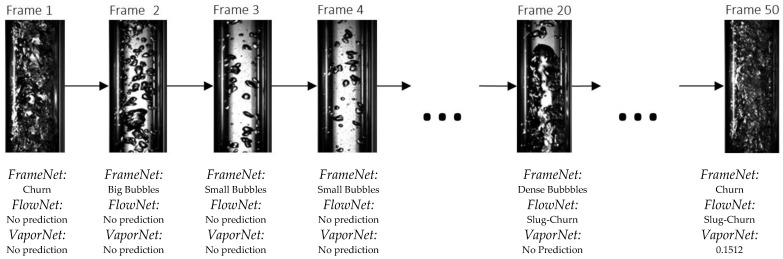
Example of model predictions across a sequence of frames.

**Table 1 sensors-22-00996-t001:** The training hyperparameters.

Network	FrameNet	FlowNet	VaporNet
Optimizer	SGD	*Adam*	*Adam*
Learning Rate	1 × 10^−3^	1 × 10^−4^	1 × 10^−4^
Batch Size	10	256	256
Training Epochs	30	60	100
Momentum	0.9	Adaptive	Adaptive

**Table 2 sensors-22-00996-t002:** Findings of the 5-fold test results, with best results in bold.

Fold	FrameNet Accuracy (%)	FlowNet Accuracy (%)	VaporNet RMSE in Vapor Quality Prediction
Fold-1	**93.0**	91.5	4.8 × 10^−2^
Fold-2	92.5	**95.4**	5.2 × 10^−2^
Fold-3	91.0	92.5	**4.4 × 10^−2^**
Fold-4	90.6	87.4	6.1 × 10^−2^
Fold-5	92.3	91.8	6.4 × 10^−2^
Mean	91.9	91.7	5.5 × 10^−2^
Standard Deviation	0.9	2.6	0.8 × 10^−2^

## Data Availability

Data available on request due to restrictions of privacy.

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
