# Peer review of "Computer Vision-Based Classification of Flow Regime and Vapor Quality in Vertical Two-Phase Flow"

_sensors, 2022, doi:10.3390/s22030996_

Round 1

Reviewer 1 Report

This paper presents a method to classify flow regime and vapor quality in vertical two-phase (vapor-liquid) flow using video of the flow as the input. The approach makes use of computer vision techniques and deep learning to train a convolutional neural network (CNN), which is used for individual frame classification and image feature extraction. A deep long short-term memory (LSTM) network is used to capture temporal information present in a sequence of image feature sets, and to make a final vapor quality or flow regime classification. The work is interesting and meaningful. And the reviewer has some questions as follow:

  1. What is the innovation of this paper? As far as I know, Convolutional Neural Networks (CNNs) have been successfully applied to two-phase flow pattern recognition and published many relevant results.
  2. Will the training sample size affect the accuracy of flow pattern judgment? If so, how?
  3. Figures 3 and 4 are not clear and should be replaced by pictures with obvious features and clear background.
  4. What are the advantages of the proposed method compared with the existing similar methods?
  5. The information shown in Figure 5 is not clear.
  6. Part 4.1 points out that there is confusion in the identification of Churn and Annular flow. So, next, does the author have any plans to improve this problem?
  7. Section 4.1 points out that the confidence interval for the prediction of slug flow steam quality is low. Is this related to the training sample size? Or is it related to the extracted flow pattern characteristics?
  8. For some flow patterns, such as mist flow, the classification accuracy decreases. Is this related to the training sample size? Because the distinguishing characteristics of each flow pattern are still obvious through visual images.

Author Response

Dear Reviewer,

Thank you for your insightful and well-posed comments. Please see below the authors responses to your points:

  1. In the revised version of the paper, lines 72-95 highlight the novelty and benefits presented in the paper. Lines 61-70 detail the most similar experiments to this one (which also make use of CNNs), but the novelty of this work in terms of its implementation of CNNs for multiphase flow studies can be found by comparing those lines with lines 72-95.
  2. It is common with machine learning experiments to train models with as much data as available to the scientist. Because of the heuristics involved in selecting training sample sizes, it is difficult to say with any confidence the effect of training the models with more data. It is commonly accepted in the world of machine learning that the use of more data during training results in better performing models, but as said, it is difficult to make claims such as these, especially with novel data as is found in the current work. Because of these points, the authors feel that no strong claims can be made regarding what the effect would be if the models were trained with more data.
  3. In the revised work, Figure 3 has been changed to Figure 4. Because the feature detectors learnt within the kernels of the convolutional layers of a CNN are optimised by gradient descent during training, without human intervention, these "features" will not necessarily be intuitive for a human to understand. This is in contrast to hand-engineered feature detectors found in traditional image processing research, which are developed to detect specific textures or geometries within an image. The features extracted from the input image by a CNN might be difficult for a reader to interpret, but they accurately display the reduction of data performed by the network. The purpose of this figure is not for the reader to follow how features are manipulated by the network, but rather to show how the network reduces a large number of input features to a single dense feature layer. In the authors' opinion, this image is appropriate and should remain unchanged. Regarding Figure 4, now Figure 5, the reviewer's comments are well taken, and the images have been cropped and enlarged to highlight the flow patterns within the images.
  4. The advantages of the proposed methods have been highlighted in lines 80-87 and lines 91-95 in the revised version of the paper.
  5. This figure has been split into two figures (6 and 7) to enlarge the individual figures and be more readable. 
  6. A suggestion for solving this problem is presented from lines 558-561. As stated, this is a challenging problem, out of scope for the current work, but an attractive topic for further research.
  7. VaporNet makes use of a balanced dataset (as stated in lines 365-366), and the model performs well for other vapor quality ranges. This implies that the dataset is sufficient in training the model to classify specific vapor quality ranges accurately, yet the range in question remains a challenge. This implies that it is more likely the features or characteristics of this vapor quality range which are the source of the uncertainty and not the size of the dataset. These ideas are expressed in lines 367-374. The authors' specific explanation for the cause of this drop in accuracy for this vapor quality range is stated in lines 544-549.
  8. The accuracy of the Mist flow classification is only poor for the real-time prototype, with an explanation for this drop in accuracy expressed in lines 597-606. This accounts for the drop in accuracy seen in figure 7(b). Regarding accuracy drops in Figure 7(a), because the network performs very well on certain regimes, and the dataset used was balanced (as stated in lines 365-366), the data should therefore be sufficient to train the network to classify a specific flow regime, as is expressed in lines 367-374. The accuracy drops are therefore more likely the result of specific characteristics found within the different regimes, being highlighted in lines 504-510 and 517-526.

Once again, the authors would like to thank you for your time and insights in reviewing this work.

Best regards,

Shai Kadish

Reviewer 2 Report

Summary

This article has a good layout for reading, and the proposed method is very interesting and inspiring. The authors proposed an image- and image sequence-based flow regime and vapor quality identification method, which achieves excellent performance through segmented training and deep learning strategies. The proposed architecture can be separated into three parts, FrameNet, FlowNet, and VaporNet. The training purpose of FrameNet is to extract high-level features (e.g., liquid, bubble size, bubble density, etc.), and its convolutional networks can be used as an automatic feature extractor. FlowNet mainly applied the LSTM network, while the LSTM has an essential challenge of balancing the dimension-reduction and information loss. However, the challenge has been solved by using the CNN in the FrameNet, which prevents the LSTM from facing a high dimension tensor (this is a very clever idea). VaporNet is a parallel network with FlowNet, using DNN for classification. The method part has been discussed in detail, and the result analysis is complete and systematic. In general, this article is a good research.

A few minor descriptions and format issues require to be corrected. Furthermore, the article needs to highlight its novelties and technology contributions, and it can be either a quantitative discussion or a qualitative comparison.

Minor issues

  1. Line 31-40:

The discussion is good to follow and comprehensive, while some reference is necessary to support these discussions.

For example, Line 37 mentioned, “Much of the literature …”, it will be good to list a few of them.

  1. Reference (include but not limited to) [1, 9, 16, 30, 31, 32, 38]: Please double check the format for all references, which contains many errors.

I would suggest the authors use professional software for managing the reference format (e.g., Mendeley). It might help to solve the issue.

e.g., the first and last names, the spelling, time, etc.

  1. Line 41-44:

The authors mentioned the “direct measurement” and “non-intrusive manner,” but the authors did not explain why the image-based technology is necessary. Or, in other words, there are many existing methods, but why is the proposed technology is beneficial or essential? This is essential for clarifying the impact and novelty of this research.

  1. Line 52-58, 73-76, 119-131:

The only intuitive discussion of novelty is Line 52, which is not enough for related studies' discussion. There are two ways of highlighting the novelties.

First, a qualitative discussion of what advantages of the proposed technique compared to related studies. For example, the following studies also used non-intrusive technologies and CNN, and they have comparative (even better) accuracy. Then, why is this research unique and valuable? Line 191-131 shows some flavor of the answer, but a more targeted discussion can significantly highlight the technological contributions.

  • Nnabuife, S.G., Kuang, B., Whidborne, J.F. and Rana, Z.A., 2021. Development of Gas-Liquid Flow Regimes Identification Using a Noninvasive Ultrasonic Sensor, Belt-Shape Features, and Convolutional Neural Network in an S-Shaped Riser. IEEE Transactions on Cybernetics.
  • Zhang, Y., Azman, A.N., Xu, K.W., Kang, C. and Kim, H.B., 2020. Two-phase flow regime identification based on the liquid-phase velocity information and machine learning. Experiments in Fluids, 61(10), pp.1-16.
  • Brantson, E.T., Abdulkadir, M., Akwensi, P.H., Osei, H., Appiah, T.F., Assie, K.R. and Samuel, S., 2022. Gas-liquid vertical pipe flow patterns convolutional neural network classification using experimental advanced wire mesh sensor images. Journal of Natural Gas Science and Engineering, p.104406.

Second, a quantitative comparison to related studies is a more solid way, which might involve some re-implementation of the related studies, then test using the same data. But, it also indicates more work and time.  

  1. Line 59:

The authors mentioned Figure 2 before Figure 1, which is not a professional figure indexing.

  1. Line 59-72:

The author should double-check this paragraph, and it seems not suitable for “Introduction.” It explains the proposed architecture, which should usually be explained in the method part.

  1. Line 73-76:

I understand the authors are trying to introduce the technological contributions in this paragraph.

  1. Line 99-102:

The authors used an ambiguous way to describe lower fps “might” cause challenges for LSTM finding the correlation between frames. It is not recommended to use vague descriptions. I recommend two ways of solving the issue:

First, conduct additional experiments with lower fps, and compare their results with current results. Ideally, the fps changes should have a kind of logic instead of arbitrarily numbers (e.g., 60, 30, 15, 10, 5, etc.) Although Table 3 compared the performance among 10, 20, and 30 fps, the FrameNet architectures are different. The FrameNet is the first part of the entire architecture, and it is difficult to defend that row-2 is comparable to row-1 and row-3 in Table 3.

Second, I don’t think the influence of the frame rate on the LSTM is the focus of this research. Thus, it is easier to describe it as an experimental setting instead of discussing its influence.

  1. The font size is too tiny in Figures 5 and 7. The priority for any figure is clear, and the authors may find a way to layout the figure properly and increase the font size in Figures 5 and 7.

Author Response

Dear Reviewer,

Thank you for your insightful and well-posed comments. Please see below the authors responses to your points:

  1. The reviewer's comments are well taken, and the references used have been added in lines 38-48.
  2. The reviewer's comments are well taken, and the references have been updated and reviewed.
  3. The benefits of the camera-based approach used here are highlighted in lines 80-87 in the revised version of the paper.
  4. The reviewer's comments are well taken, and lines 61-71 were updated to compare this work with similar work done in the past. Additionally, lines 73-79 and 83-90 highlight the novelties presented in this work.
  5. The issue has been amended.
  6. The paragraph in question has been removed and replaced with a discussion of novelty and benefits, found in lines 72-95. The authors agree that this section is out of place for the introduction.
  7. The technological contribution of the paper is clarified in lines 27-95.
  8. Because the insights presented in this paper regarding the effect of FPS in video-based multi-phase flow studies are useful for practical application, it is in the opinion of the authors that the sections describing the effect of frame rate should stay in the paper in their current form. Figure 1 has been included to clarify the effect of adjusting the FPS for both ResNet101 and ResNet18. This new figure highlights the point discussed in line 578-596, which shows that frame rate is a key parameter when designing a network for studying multi-phase flow videos. This figure and lines 187-194 should remove any ambiguity as to what a "high" or "low" framerate means in this case.
  9. These figures have been adjusted. The images in the original figure 5 are now larger and displayed in figures 6 and 7. The original Figure 7 has been adjusted to have clearer text and images, and is now shown in Figure 9.

Once again, the authors would like to thank you for your time and insights in reviewing this work.

Best regards,

Shai Kadish